# Generative Expressive Conversational Speech Synthesis

Rui Liu
imucslr@imu.edu.cn
Inner Mongolia University
Hohhot, China

Yifan Hu
22309013@mail.imu.edu.cn
Inner Mongolia University
Hohhot, China

Yi Ren
ren.yi@bytedance.com
ByteDance
Singapore, Singapore

Xiang Yin
yinxiang.stephen@bytedance.com
ByteDance
Shanghai, China

Haizhou Li
haizhouli@cuhk.edu.cn
SRIBD, School of Data Science
The Chinese University of Hong Kong
Shenzhen, China
National University of Singapore
Singapore, Singapore

## Abstract

Conversational Speech Synthesis (CSS) aims to express a target utterance with the proper speaking style in a user-agent conversation setting. Existing CSS methods employ effective multi-modal context modeling techniques to achieve empathy understanding and expression. However, they often need to design complex network architectures and meticulously optimize the modules within them. In addition, due to the limitations of small-scale datasets containing scripted recording styles, they often fail to simulate real natural conversational styles. To address the above issues, we propose a novel generative expressive CSS system, termed **GPT-Talker**. We transform the multimodal information of the multi-turn dialogue history into discrete token sequences and seamlessly integrate them to form a comprehensive user-agent dialogue context. Leveraging the power of GPT, we predict the token sequence, that includes both semantic and style knowledge, of response for the agent. After that, the expressive conversational speech is synthesized by the conversation-enriched VITS to deliver feedback to the user. Furthermore, we propose a large-scale Natural CSS Dataset called **NCSSD**, that includes both naturally recorded conversational speech in improvised styles and dialogues extracted from TV shows. It encompasses both Chinese and English languages, with a total duration of 236 hours. We conducted comprehensive experiments on the reliability of the NCSSD and the effectiveness of our GPT-Talker. Both subjective and objective evaluations demonstrate that our model outperforms other state-of-the-art CSS systems significantly in terms of naturalness and expressiveness. *The Code, Dataset, and Pre-trained Model are available at: https://github.com/AI-S2-Lab/GPT-Talker.*

## CCS Concepts

• **Information systems → Multimedia content creation**.

## Keywords

Conversational Speech Synthesis (CSS), User-agent Conversation, GPT, Expressiveness

**ACM Reference Format:**
Rui Liu, Yifan Hu, Yi Ren, Xiang Yin, and Haizhou Li. 2024. Generative Expressive Conversational Speech Synthesis. In *Proceedings of the 32nd ACM International Conference on Multimedia (MM '24), October 28-November 1, 2024, Melbourne, VIC, Australia.* ACM, New York, NY, USA, 10 pages. https://doi.org/10.1145/3664647.3681697

## 1 Introduction

Conversational Speech Synthesis (CSS) (or Conversational Text-to-Speech (TTS), CTTS) aims to generate speech with proper style in the user-agent conversation scenario. In such scenarios, the user usually initiates a dialogue, then the agent and the user take turns to speak. During the interaction, the agent is expected to understand the user's needs and provide assistance and emotional support. Currently, with the increasing popularity of smart devices, there is a growing demand for human-machine interaction in various application scenarios such as smartphone assistants [47], smart home control [21], intelligent vehicle systems [15], and virtual reality / augmented reality interactions [13].

Many attempts on CSS have been proposed to enhance the naturalness and expressiveness of synthesized conversational speech from the perspective of context modeling. Guo et al, [17] proposed a GRU-based context encoder to extract global prosodic information for the agent from the dialogue context. FCTalker [19] further incorporates the word- and sentence-level context knowledge, that represents the fine- and coarse-grained context, to enhance the context understanding ability of the agent. However, these works only consider the textual information, ignoring the audio and multi-modal dependencies in the conversation. To this end, researchers contributed to the study of multi-modal context modeling [10, 35, 39, 40]. However, they are increasingly inclined towards designing complex network architectures and optimizing the modules within them meticulously. For example, multi-scale multi-modal CTTS ($M^2$-CTTS) system [53] includes a textual context module and an acoustic context module with both coarse-grained and fine-grained modeling. Li et al. [28, 29] and Liu et al. [34] proposed the graph neural network based context learning schemes.

With the advancement of the Large Language Model (LLM), several studies aim to construct spoken language models that extend language models for the speech domain [23]. We have noticed that the Generative Pre-training Transformer (GPT) possesses concise and powerful context modeling capabilities [14] and has demonstrated impressive performance in capturing fine- and coarse-grained dependencies in tasks such as dialogue generation [58] and understanding [32, 33, 56]. Despite the successes, all the above works are not applicable to CSS, where multi-modal context is used as input to generate response speech output. We note that the context modeling of GPT aligns well with the requirements of CSS, yet it has been largely overlooked. Therefore, how to leverage GPT to build concise yet powerful context understanding solutions for user-agent interaction, will be the focus of this work.

In addition, the existing spoken dialogue dataset is challenging to meet the requirements for training GPT-based generative expressive CSS models in terms of both scale and quality. Specifically, $M^2$-CTTS [53], EmoSit-TTS [35] and ECSS [34], etc. all utilized a small-scale DailyTalk dataset [27], which contains 2541 dialogues audio in about 20 hours in total. More importantly, it only consists of recordings in a reading style, lacking sufficient expressiveness. Some works [10, 29] have attempted to use datasets such as IEMO-CAP [4] and ECC [28]. However, these datasets are designed for purposes like conversational emotion recognition [4, 41] and conversation education, and they may contain background noise issues that can affect the synthesis results. Therefore, it is necessary to construct a large-scale and high-quality expressive CSS dataset to support GPT-based CSS models in achieving a real natural conversational style.

In this paper, we propose a novel generative expressive CSS system, termed **GPT-Talker**. We transform the multimodal information of the dialogue history into discrete token sequences and seamlessly integrate them to form a comprehensive user-agent dialogue context. Leveraging the power of GPT, we predict the token sequence, that includes both semantic and style knowledge, of response for the agent. After that, the expressive conversational speech is synthesized by the conversation-enriched VITS to deliver feedback to the user. Furthermore, we propose a large-scale Natural CSS Dataset called **NCSSD**, that includes both naturally recorded conversational speech in improvised styles and dialogues extracted from TV shows. It encompasses both Chinese and English languages, with a total duration of 236 hours. The NCSSD dataset and related annotation will be public freely. We conducted comprehensive experiments on the reliability of the NCSSD and the effectiveness of our GPT-Talker. Both subjective and objective evaluations demonstrate that our model outperforms other state-of-the-art CSS systems significantly in terms of naturalness and expressiveness. In summary, our main contributions are as follows: 1) To the best of our knowledge, we are one of the earliest to introduce GPT into conversational speech synthesis and build a concise and powerful context modeling scheme for the user-agent conversation. 2) We have proposed a new large-scale Natural CSS dataset, termed NCSSD, that can support our GPT-Talker, even future GPT-style CSS model to achieve a real natural conversational speech style. 3) We have conducted comprehensive validations of the model's effectiveness and the dataset's reliability. Our model

significantly outperforms baseline models in terms of naturalness and expressiveness.

## 2 Related Work

### 2.1 Conversational Language Model

Text and audio are two important modalities for human communications. Text-based LLMs have demonstrated remarkable achievements across various domains, including conversational chatbots [2], code generation [6], creative writing [46], and machine translation [36].

Inspired by the aforementioned works, many studies have recently explored conversational language modeling to address a variety of tasks involving speech and text [48, 49], such as automatic speech recognition [12], spoken question answering [1], and speech-to-text translation [51], etc. For instance, SpeechGPT [57] exhibited cross-modal conversational capabilities by employing discrete unit representations to convert continuous speech signals, and integrating LLMs with unit vocoder. This enables the model to effectively process multimodal input and generate corresponding output. dGSLM [38] proposed a dual-tower spoken LLM on discrete speech units to model two-channel spoken dialogue, but the generated spoken sentences lack semantic meaning. USDM [24] proposed a generalized speech-text pretraining scheme that leverages the chain-of-reasoning capabilities of LLMs to generate coherent spoken responses based on conversational speech. In human conversation, while the dialogue primarily relies on the lexical aspect, the speaking styles convey rich information beyond text, and can even alter the semantics of the spoken sentences [5]. To this end, E-Chat [52] proposed a novel LLM-based spoken dialogue system, that leverages an emotion embedding extracted by a speech encoder, enabling it to respond according to different emotional contexts. ParalinGPT [31] takes the conversational context of text, speech embeddings, and paralinguistic attributes as input prompts for LLM to improve current and response sentiment prediction, as well as response text generation, in natural human-human speech dialogue. Furthermore, Spoken-LLM [32] proposed to fuse the LLM and a self-supervised speech emotion representation model to help the LLM to predict response speaking style and text, enabling the subsequent expressive TTS model to generate natural and diverse speech responses.

Our work performs significantly differently from the above-mentioned studies. Specifically, our work focuses on the task of CSS, where we integrate multi-turn multi-modal dialogue context into a unified sequence and predict the semantic and stylistic representations of the speech to be synthesized along with the current utterance. We then utilize this representation to generate the final expressive conversational speech. However, the aforementioned works, such as dGSLM, SpeechGPT, and USDM, overlooked effective mechanisms for modeling multi-turn dialogue history, while E-chat, ParalinGPT and Spoken-LLM etc. only generate text responses based on dialogue history rather than speech representations. Although MQTTS [7] and Pheme [3] also claim to be conversational speech generation systems, they do not model the dialogue context and only focus on the naturalness of the synthesized speech. Our

**Table 1: Comparison among other conversational datasets and NCSSD. ∗ means that the handling of collection data requires manual involvement. (EN: English; CN: Chinese; RC: Recording; CL: Collection.)**

| Dataset | Language | Source | Scale | |
|---|---|---|---|---|
| | | | Duration (h) | Speaker |
| IEMOCAP [4] | EN | RC | 12 | 10 |
| MELD [41] | EN | CL* | 13.66 | 407 |
| M$^3$ED [59] | CH | CL* | 14.14 | 626 |
| CPED [8] | CH | CL* | 78.34 | 392 |
| ECC [28] | EN | CL* | 24 | 26 |
| DailyTalk [27] | EN | RC | 20 | 2 |
| STUDIES [44] | JP | RC | 8.2 | 3 |
| ASLP-CH [17] | CH | RC | 3 | 2 |
| RyanSpeech [54] | EN | RC | 10 | 1 |
| CALLS [43] | JP | RC | 6.5 | 1 |
| **NCSSD (Ours)** | **EN, CH** | **RC, CL** | **236** | **>776** |

GPT-Talker is similar to the recent style transfer TTS model, GPT-SoVITS [16], but an obvious difference is that GPT-SoVITS does not pay attention to the dialogue context information.

## 2.2 Conversational Speech Datasets

Table 1 provides a summary of the existing relevant datasets to our knowledge. Lines 6-11 present information about some conversational speech datasets specifically designed for CSS task. It can be observed that these datasets are relatively small in scale. For example, ECC [28] gathered 66 sets of public videos from YouTube's English Conversation channel, amassing 24 hours of content. These dialogues involve two, three, or multiple participants. DailyTalk [27] is derived from DailyDialog dataset [30]. It showcases 2,541 high-quality English conversations between a male and female, spanning a total of 20 hours. The Japanese corpus for empathic conversations STUDIES [44] and CALLS [43] covers two scenarios, that are *communication between teachers and students in school* and *customer service via telephone*, featuring 8.2 and 6.5 hours of speech, respectively. RyanSpeech [54] is a high-quality male TTS corpus in the conversational domain, that contains 9.84 hours audio samples. Note that CALLS and RyanSpeech claim to be conversational datasets, but they only contain one speaker and are not suitable for CSS task. Guo et al. [17] recorded an internal dataset (we call "ASLP-CH" here) comprising 3 hours of conversations between 2 Chinese women, who assumed the roles of a customer and a customer service representative, respectively.

In summary, the scale of these datasets is insufficient to train a high-quality CSS model based on the GPT model. This necessitates the creation of a large-scale, freely available CSS dataset. Additionally, our dataset has advantages in terms of *language diversity* and *data source diversity* compared to existing datasets. Specifically, it includes bilingual data in both Chinese and English, as well as subsets of recorded data and collected data. Furthermore, we have developed an automated processing pipeline for collected data, greatly improving the efficiency of creating natural CSS datasets. We hope that this initiative can contribute to the development of

the community. Furthermore, in Table 1, we have listed some of the latest dialogue speech datasets designed for tasks such as dialogue emotion understanding. It can be seen that our data still has significant advantages in terms of language diversity, data source diversity, and dataset scale compared to these datasets.

## 3 GPT-Talker: Methodlogy

### 3.1 Task Definition

In user-agent spoken conversation, the user and the agent take turns speaking. The user speaks first, and the agent understands the user's semantics to provide a spoken response. As time progresses, multi-turn user-agent dialogue history accumulates and forms. The task setting in our CSS task, therefore is to generate the speech of agent's response according to the conversation context, where the text of the response is given. Specifically, assume that the $N$-turn conversation context includes the multi-modal dialogue history $\mathcal{H}$ and the current utterance $C$, where $\mathcal{H} = \{(U_1^t, U_1^a, S_1), (U_2^t, U_2^a, S_2)..., (U_{N-1}^t, U_{N-1}^a, S_{N-1})\}$ and $C = (U_N^t, S_N)$. ($t$ and $a$ means the text and audio modalities respectively, $S$ is the speaker identify label.) The goal of CSS is to ensure that the synthesized speech $U_N^S$ is suitable for the whole dialogue situation. Therefore, how to model the multi-turn multi-modal conversational context and provide a proper speaking style for the agent is the focus of the task.

To this end, our GPT-Talker consists of two key components, that are *GPT-based context modeling* and *Expressive conversational speech synthesis*. Specifically, GPT-based context modeling proposes a novel Conversation GPT (ConGPT) to model the multi-turn multi-modal conversational context by treating the discrete token sequence of context as the condition prompt, and predict proper semantic and style expression for the agent. Expressive conversational speech synthesis proposes the Conversational VITS (ConVITS) to enrich the VITS with the agent's semantics, style, and timbre to generate expressive conversational speech based on the known response content.

### 3.2 Conversational GPT

As shown on the left side of Fig. 1, The ConGPT encompasses 1) Multi-turn Multi-modal Context Tokenization and 2) ConGPT-based Semantic and Style Inference. The former module converts the multi-modal context into a unified discrete sequence, and then constructs a discrete representation of the multi-turn conversation context. Based on this representation, The latter module infers the semantics and style of the response speech.

*3.2.1 Multi-turn Multi-modal Context Tokenization.* Unlike traditional CSS works that adopt complex graph neural networks [29, 34] or cascade pipelines [35] to understand the context, our ConGPT seeks to understand the multi-modal context within a unified discrete sequence directly.

**Textual Tokenization**: Similar with VALL-E [48], we convert the text data of dialogue history $\mathcal{H}$ and current utterance $C$, including $U_{1\rightarrow N-1}^t$ and $U_N^t$, into the discrete phoneme sequences. As shown in Fig. 1, the discrete phoneme sequences of $U_{1\rightarrow N}^t$ are represented by $T_{1\rightarrow N}^t$. The phoneme encoder is built based on the g2p_en

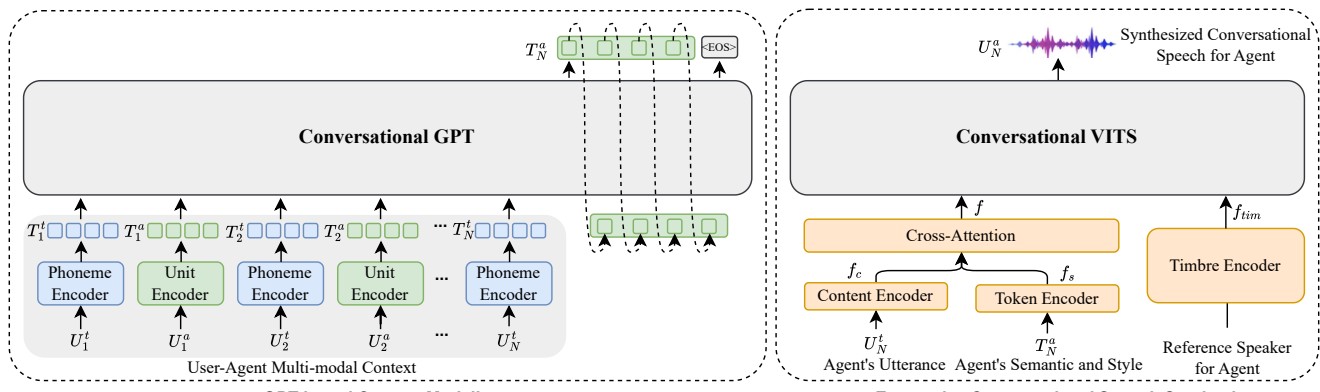

**Figure 1: The overview of GPT-Talker, that includes the Conversational GPT and the Conversational VITS.**

python module [1] for English and the opencpop-strict dictionary [2] for Chinese.

**Acoustic Tokenization**: To model natural speech conversations, the discrete speech representation must contain not only the semantics of the utterance but also expressiveness features such as speaking style, which are crucial for CSS task. Previous research [24] has demonstrated that HuBERT tokens contain rich semantic information as well as strong traces of paralinguistic features that can be used to accurate classify speech emotions. Therefore, we follow [24] and employ HuBERT [18] to acquire high-dimensional speech representations. Inspired by [26], we do not employ unit deduplication on HuBERT units, since it maintains a consistent length ratio with the speech, removing the need to prepare a separate duration predictor. Specifically, we adopt speech units as acoustic context units that are derived from k-means clustering of the HuBERT's intermediate speech representation. We follow GPT-SoVITS and then adopt a Vector Quantization (VQ) layer to convert the acoustic context units to the learnable token sequence, thus better matching the expressive speech generation task. In this way, the audio data of dialogue context $U_{1 \to N-1}^a$ are represented by $T_{1 \to N-1}^a$.

**Unified Context Serialization**: After acquiring discrete tokens for text and audio modalities of context, we emulate the real-time dialogue flow and combine the textual and acoustic tokens alternately into a multi-modal conversational format as $\{T_1^t, T_1^a, T_2^t, T_2^a, ..., T_N^t\}$. We adopt an alternating approach between the text and speech modalities instead of serializing the conversation history by first combining the text modality and then concatenating the speech modality. We believe that the alternating approach can better capture the cross-modal contextual dependencies in the conversation history.

*3.2.2 ConGPT-based Semantic and Style Inference.* To inference the Semantic and Style information of response speech according to the serialized multi-modal user-agent context. the ConGPT takes the entire serialized conversation context $\{T_1^t, T_1^a ..., T_{N-1}^t, T_{N-1}^a, T_N^t\}$ as input, then predicts the semantic and style knowledge $T_N^a$. The process is formulated as follows:

$$T_N^a = ConGPT(\{T_1^t, T_1^a, T_2^t, T_2^a ..., T_N^t\}) \tag{1}$$

---

[1]https://pypi.org/project/g2p-en/
[2]https://wenet.org.cn/opencpop/

where $T_N^a$ also follows the same tokenization method as $U_{1 \to N-1}^a$ to discretize $U_N^a$ in advance, serving as the output of ConGPT. The sub-tokens in the discrete token sequence $T_N^a$ are decoded using an autoregressive approach until the End of Sequence (EOS) label is decoded. Please note that, similar to [33], we do not assign explicit speaker identities to the token sequence of the serialized context.

### 3.3 Conversational VITS

After unified context modeling, the Conversational VITS (ConVITS) is proposed to provide proper expressive speech for the agent's response. Note that ConVITS takes three information sources of the agent, including the agent's utterance $U_N^t$ for content rendering, the semantic and style representations of the agent inferred by the ConGPT for semantic and style rendering, and the additional reference speaker for timbre rendering. As shown in the right panel of Fig. 1, it consists of four key complements, which are the context encoder, token encoder, timbre encoder and the VITS synthesizer. The cross-attention mechanism is used to integrate the content, semantic and style information of agent.

*3.3.1 Content, Semantic and Style Rendering.* First, we utilize the text encoder $Encoder_t(\cdot)$, that consists of 6 layers of transformer encoder, to process the response content $U_N^t$, extracting its inherent textual information $f_c$ to ensure the intelligible of synthesized speech. After that, the predicted context-aware semantic and style tokens $T_N^a$ are converted into a high-level style representation $f_s$ using the token encoder $Encoder_s(\cdot)$, which consists of 3 layers of transformer encoder.

Finally, to achieve unified content, semantic and style rendering for expressive CSS, textual information $f_c$ and high-level style representation $f_s$ are integrated into a final agent's style embedding $f$ via the cross-attention layer. We use a multi-head attention module in the cross-attention layer, which accommodates different input lengths. Here, we treat $f_s$ as the query and $f_c$ as both the key and value.

*3.3.2 Timbre Rendering.* In order to achieve flexible and diverse expressive styles, we do not rely on fixed speaker IDs for speaker control. Instead, we employ a timbre encoder with a reference encoder to perform timbre transfer on any reference speech, allowing for flexible speaker timbre rendering based on content, semantics, and style rendering. As shown in the Fig. 1, the timbre encoder

$Encoder_{tim}(\cdot)$, consists of 6 convolutional layers, a GRU layer, and a linear layer, takes a reference speech $U^a_{agent}$ for a specific speaker and extracts the timbre embedding $f_{tim}$.

During training, the speech $U^a_{N-2}$ from the agent in the last turn is chosen as the reference speech $U^a_{agent}$. During the inference stage, we can assign a reference with any speakers to achieve zero-shot timbre rendering performance.

*3.3.3 Speech Generation.* The speech generation stage extends the vanilla VITS [25] into conversational settings by leveraging the above conversational expressiveness rendering. It's important to highlight that the ConVITS architecture excludes the Stochastic Duration Predictor, since the semantic and style tokens predicted by ConGPT already encompass duration information, there's no necessity to predict it separately.

The final agent's style embedding $f$ is processed by a projection layer to derive the mean $\mu$ and variance $\theta$. The agent's timbre embedding $f_{tim}$ is encoded by the posterior encoder to produce the latent normal posterior distribution variable $z$, which is then decoded by a Flow-based Decoder to generate the normalized stream $f_\theta(z)$. The HiFi-GAN generator then upsamples the latent variables $z$ to the speech waveform $U^a_N$.

## 3.4 Training Strategy

We propose a **Three-Stage** training strategy to ensure the performance of our GPT-Talker: 1) In the first stage, we focused on the modeling capabilities of ConGPT and ConVITS in single-sentence speech scenarios. we trained ConGPT and ConVITS with single-sentence speech datasets including LibriTTS [55], LJSpeech [20], AISHELL-3 [45], etc., that the total duration is about 2.5k hours. In this way, ConGPT can predict the speech tokens based on the text token sequence. The ConVITS performs stable speech generation after inputting the provided text. 2) In the second stage, we continue to train the ConGPT using the collection subset of NCSSD. This enabled it to accurately predict the semantics and stylistic knowledge of the current sentence, leveraging the provided dialogue context and multimodal information. 3) In the third stage, we further enhance the naturalness and expressiveness of the synthesized speech by fine-tuning both ConGPT and ConVITS using the recording subset of NCSSD.

Concerning the computation of the model's total loss. The GPT-Talker's loss $L_{total}$ is composed of two elements: $L_{ConGPT}$ and $L_{ConVITS}$. $L_{ConGPT}$ calculates the cross-entropy loss between the predicted acoustic units and the real units. $L_{ConVITS}$ includes mel reconstruction loss $L_{recon}$, KL divergence loss $L_{vae}$ from VAE, feature-matching loss $L_{fm}$, adversarial training loss $L_{adv}$, and $L_{vq}$ from Vector Quantization. Specifically, $L_{vq}$ computes the commitment loss between the quantized vector and the input discrete token.

## 4 NCSSD Construction

As mentioned before, we also propose a large-scale natural conversational speech corpus, **NCSSD**, to support the GPT-based CSS training. As illustrated in Table 1, the dataset includes two subsets, that are the collection part and the recording part, covering a total duration of over **236 hours**. Our dataset is available with the CC-BY-SA 4.0 license. Unlike the traditional data production method with human participation [8, 41, 59], note that we propose

an automatic data construction pipeline for the collection subset and a ChatGPT-assisted workflow for the recording subset. We will report the data collection process [3] and the data statistics results.

### 4.1 Automatic Data Construction for Collection Subset

The automatic data construction pipeline for the collection subset consists of 1) Video Selection, 2) Dialogue Scene Extraction, 3) Dialogue Segment Extraction, and 4) Dialogue Script Recognition.

*4.1.1 Video Selection.* In order to build a large-scale, diversified, and natural conversational dataset, we collect videos from different TV series, which can simulate spontaneous conversation behavior in the real-world environment [41, 59]. We collect 79, and 34 TV shows for English and Chinese respectively. The detailed list of all TV series are introduced in Appendix. Unlike other data creation methods that require manual selection of TV shows based on certain rules [59], we automatically filter out eligible dialogues and corresponding timestamps using *Dialogue Scene Extraction*, *Dialogue Segment Extraction*, and *Dialogue Script Recognition* modules.

*4.1.2 Dialogue Scene Extraction.* A complete TV show consists of multiple dialogue scenes (may include two or more speakers) that are interconnected but independent from each other. To extract these dialogue scenes, we employ Voice Activity Detection (VAD) technology, which uses silent segments in the audio information of the entire TV show to identify dialogue scenes. Subsequently, the extracted dialogue scene audio is further processed to obtain clean dialogue speech.

Specifically, we first employ a pre-trained VAD model, silero-vad[4], to identify the timestamps of non-silent voice chunks in the video, since the silero-vad was trained on huge corpora that include over 100 languages and it performs well on audios from different domains with various background noise and quality levels. We set the silent segments threshold to 4 seconds to get the VAD results, since 4 seconds of silence often indicates the start of a new dialogue, as proven by extensive preliminary data testing. Then we segment the complete audio file of a video into various discontinuous audio clips that represent the various dialogue scenes. To ensure that each dialogue scene includes multi-turn dialogues with appropriate length, we further discard the audio clips where "the ratio of silent to non-silent segments exceeds 30%" or the "total duration is less than 15 seconds".

Subsequently, we then perform background music separation with Demucs [5] to discard the background music and other distracting information. To further refine the vocal component's quality, we thoroughly assess the signal-to-noise ratios (SNR) of both vocal and background noise, retaining only vocal chunks with an SNR above 4, and adopt speech enhancement model, sepformer [6], to obtain the clean audio for all dialogue scenes.

*4.1.3 Dialogue Segment Extraction.* Dialogue segment extraction aims to extract the conversational speech containing only two-person interaction from the previously obtained dialogue scene speech.

---

[3]For a more intuitive flowchart, please refer to the Appendix.
[4]https://github.com/snakers4/silero-vad
[5]https://github.com/facebookresearch/demucs
[6]https://huggingface.co/speechbrain/sepformer-dns4-16k-enhancement

To achieve this, we utilized a speaker recognition interface [7] based on ByteDance to analyze the speech information from all dialogue scenes. This interface provides us with numerical speaker labels and corresponding timestamp information. Subsequently, we extract dialogue segments for two-person conversations based on the speaker labels and timestamps. Our extraction criteria required the dialogue segment to have more than four utterances, with each speaker contributing at least two utterances.

*4.1.4 Dialogue Script Recognition.* Through the above steps, we obtained the dialogue-level audio and visual information of the two-person conversation. In order to obtain the dialogue script, we employ the Alibaba automatic speech recognition engine [8] to recognize the speech. Finally, the audio, scripts and the visual information of all dialogues are combined tighter as the collection suset of NCSSD.

## 4.2 ChatGPT-assisted Data Construction for Recording Subset

Unlike the collection subset, the construction of the recording subset involves designing dialogue scripts and inviting volunteers to record their voices and upper body image signals. The script serves as a prompt for the dialogue content rather than a strict guideline, and volunteers are allowed to spontaneously expand on the dialogue during the recording process. The resulting dialogue speech, the upper body image and the script are then recorded to obtain the final recorded data.

*4.2.1 Dialogue Script Draft Generation.* The first step in preparing recorded speech data is to create dialogue scripts. Manual preparation of scripts can be time-consuming, labor-intensive, and may lack diversity in terms of dialogue topics. Therefore, we leverage the powerful text generation capabilities of ChatGPT by using a prompt-based approach to generate diverse dialogue scripts that meet our expectations.

Specifically, we employ GPT-3.5 Turbo version of ChatGPT to generate the script of a two-person conversation. As shown in Fig. ??, we design a two-step prompt template to prompt the ChatGPT to output large-scale dialogue scripts to reflect spontaneous conversation behavior in the real world. In the first step, we aim to prompt ChatGPT to generate various topic words of human conversation ensuring the richness of communication content. In the second step, we select a specific dialogue topic and set the emotional state of the starting speaker. We prompt ChatGPT to generate multi-turn dialogues with a range of 4 to 15 turns. Additionally, we request GPT to add emotion and intent labels [9] to the generated scripts to enhance their interpretability and provide more reference information for subsequent speech recordings. The detailed prompt templates are shown in Appendix.

*4.2.2 Spoken Dialogue Recording.* Based on the previously obtained scripts, we invited volunteers to record the dialogue speech. We also captured the upper body images of the participants during

**Table 2: The detailed data statistics of NCSSD.**

| Item | Collection | | Recording | |
|---|---|---|---|---|
| | EN | ZH | EN | ZH |
| Language | *EN* | *ZH* | *EN* | *ZH* |
| Dialogues | 7,033 | 8,776 | 1,196 | 2,451 |
| Utterances | 62,603 | 99,126 | 10,033 | 21,688 |
| Words | 856,011 | 1,688,778 | 157,967 | 507,008 |
| Min words per utterance | 1 | 1 | 2 | 2 |
| Max words per utterance | 299 | 322 | 53 | 92 |
| Duration(h) | 72.94 | 115.22 | 19.10 | 29.57 |
| Max dialogue duration(s) | 177.51 | 308.21 | 108.52 | 75.82 |
| Min dialogue duration(s) | 4.74 | 5.54 | 20.06 | 15.73 |
| Mean utterance duration(s) | 4.19 | 4.18 | 6.85 | 4.90 |
| Max dialogue turns | 39 | 69 | 15 | 14 |
| Mean dialogue turns | 8.90 | 11.29 | 8.38 | 8.84 |
| Min dialogue turns | 4 | 4 | 5 | 6 |
| Speakers | > 339 | > 410 | 11 | 16 |

the recording. Importantly, during the recording process, volunteers are encouraged to freely add dialogue content based on the emotional and intent information provided in the script, in order to create spontaneous and natural dialogue speech.

Specifically, we employed 27 young volunteers with English as their second language [10] to participate in the recording sessions. Their compensation is based on the number of dialogues recorded. The recording venues varied, including classrooms, meeting rooms, seminar halls, and more, providing a diverse range of settings. We allow the volunteers to engage in spontaneous dialogues guided by the provided script, emotion label, and intent label. This approach ensured that the final recorded voice sounded natural and authentic.

*4.2.3 Dialogue Script Re-identification.* Due to the presence of spontaneous utterances by the voice actors during the recording, we use a speech recognition interface [8] to re-transcribe the recorded speech and obtain the recognized text as the final dialogue script. Together with the recorded speech and upper body images, this forms our final recording subset.

## 4.3 Data Statistics

Table 2 presents the overall statistics of the NCSSD dataset. It contains a total of 19,456 dialogues and 193,450 sentences from 113 TV shows, ensuring the scale and diversity of the data. The dataset spans approximately 236 hours, featuring the longest Chinese conversation at 308.21 seconds and the longest English conversation at 177.51 seconds. Every conversation exceeds 4 seconds, satisfying the minimum duration needed for dialogue tasks. On average, each conversation contains at least 8 turns, facilitating the training of models for extended dialogue sequences. With over 776 speakers, the dataset encompasses a diverse range of speaking styles and habits. For additional statistics of the NCSSD, please refer to Appendix.

---

[7]https://www.volcengine.com/product/voice-tech

[8]https://ai.aliyun.com/nls/

[9]The emotion categories are labeled using a 7-category scheme [59], that are Neutral, Happy, Surprise, Fear, Angry, Disgust, and Sad. The intention labels are labeled using a 9-category scheme [50], including Question, Agree, Acknowledge, Sympathize, Encourage, Console, Suggest, Wish, and Neutral.

[10]Due to budget limitations, we did not invite native English speakers for the recording. Although the language proficiency of our participants may not be their native language, they possess fluent English listening, speaking, reading, and writing skills. We believe they are competent enough to carry out our data recording tasks.

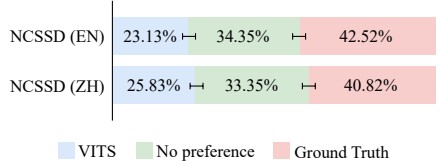

**Figure 2: The ABX preference test results between VITS and Ground Truth (GT) speech on our NCSSD.**

## 5 Experiments

In the experiment part, we will first assess the quality of NCSSD in a single-sentence speech synthesis scenario. Subsequently, we validate the GPT-Talker by comparing it with various state-of-the-art baselines of CSS task. We will introduce the baselines, and evaluation metrics in the following subsections and report the results in the next section. Note that the experimental setup can be viewed in the Appendix.

### 5.1 Baseline Models

To validate the reliability of NCSSD, we evaluate it on the single-sentence speech synthesis task and select the advanced end-to-end TTS model, VITS [25], as the baseline. Note that VITS was also chosen as the backbone network by many CSS models [9, 11], and we believe it is appropriate to serve as the baseline.

To evaluate the validity of GPT-Talker in terms of context understanding and modeling, we develop three advanced CSS systems that represent various context learning methods with the following three categories: (1) **GRU-based Context Learning**: The conversational context-aware TTS (we call "CCATTS" here) model proposed by [17] employs a GRU-based network to model the sentence-level dependency among the dialogue context; (2) **Multi-scale Context Learning**: FCTalker [19] is an representative work that consider both the sentence-level and word-level contextual within the dialogue context; (3) **Heterogeneous Graph-based Context Learning**: ECSS [34] is an advanced expressive and emotional CSS model that adopts heterogeneous graph to model the complex relation among the multi-modal context.

### 5.2 Metrics

We conduct comprehensive evaluations using both objective and subjective metrics.

**Subjective Metrics.** The subjective evaluation includes three components: (1) **ABX preference test**, where listeners had to decide which of the transformed speeches, A or B (produced by two different methods), sounded closer to ground truth speech or if they had no preference. (2) **Dialogue-level Mean Opinion Score in terms of Naturalness (N-DMOS)**, which requires participants to primarily assess its naturalness under the dialogue context. (3) **Dialogue-level Mean Opinion Score in terms of Emotion (E-DMOS)**, which primarily focuses on assessing the emotional expression conveyed through the speech and determining its alignment with the ongoing emotional tone of the dialogue context.

**Objective Metrics.** (1) Dynamic Time Warping Distance **DTWD**: we follow [42] and adopt average dynamic time warping (DTW) distance [37] of the pitch distribution for the ground-truth speech

and synthesized speech to evaluate expressiveness-related performance. (2) **Speaker Similarity (SSIM)**, we follow [22] and utilize the speaker verification model, that fine-tuned with WavLM[11], to extract the speaker embedding for the ground-truth speech and the synthesized speech. After that, we calculate the cosine similarity score between two embeddings as the final speaker similarity metric. The similarity score ranges from [-1, 1], where a higher value indicates a greater similarity.

## 6 Results and Discussions

We report the results of the following aspects, including "Reliability verification of NCSSD", "Validity verification of GPT-Talker" and "Analysis of Three-Stage Training". Please refer to the Appendix to check more results.

### 6.1 Reliability Verification of NCSSD

We train VITS using our NCSSD in single-sentence scenarios, and the ABX preference test results of synthesizing Chinese and English speech are shown in Fig. 2.

For each language, we combine the collection and recording subsets to train the VITS model. We randomly select 50 sentences from both the Chinese and English datasets for synthesis and conduct an ABX preference test with 30 volunteers to mark the speech that has higher naturalness between the speech synthesized by the VITS model and the ground truth (GT) speech. The results of Fig. 2 indicate that the performance of VITS is very close to that of the GT. The proportion of "No Preference" in Chinese speech reaches as high as 33.35%, while in English speech, it is approximately 34.35%. This demonstrates that our NCSSD can support advanced TTS model in synthesizing high-quality speech.

Undoubtedly, the performance of GT speech is expected to be better. However, since our NCSSD is designed to mimic real-world human conversations, effectively incorporating contextual modeling methods is an effective approach to further enhance the performance of TTS models. Therefore, in the next section, we further validate the effectiveness of the proposed GPT-Talker in context learning.

### 6.2 Validity Verification of GPT-Talker

In this section, we will compare GPT-Talker with three advanced CSS baselines. Since these baselines have conducted experiments using the English dataset DailyTalk in their respective studies, we will validate our model using the English portion of the NCSSD dataset and the DailyTalk dataset [12]. Similar to the previous section, the English portion will be trained by combining the collection subset and the recording subset.

For subjective evaluation, due to budget constraints, we don't invite native English speakers as volunteers. Instead, we invite 30 Chinese students who are proficient in English listening, speaking, reading, and writing as volunteers. We randomly select 50 sentences from the test set to synthesize speech for each system to be tested. Then, we ask the volunteers to rate the speech using N-DMOS

---

[11]https://huggingface.co/microsoft/wavlm-base-plus-sv

[12]Please note that directly training the GPT-Talker model on the DailyTalk dataset leads to poor performance. Therefore, the experimental results in this section are obtained by fine-tuning all systems based on the pre-training described in section 3.4.

**Table 3: Subjective and objective experimental results on DailyTalk and NCSSD datasets (CL-EN & RC-EN).**

| Methods | DailyTalk | | | | NCSSD (EN) | | | |
|---|---|---|---|---|---|---|---|---|
| | SSIM (↑) | DTWD (↓) | N-DMOS (↑) | E-DMOS (↑) | SSIM (↑) | DTWD (↓) | N-DMOS (↑) | E-DMOS (↑) |
| CCATTS [17] | 0.734 | 67.376 | 3.402 ± 0.025 | 3.429 ± 0.019 | 0.752 | 65.234 | 3.425 ±0.025 | 3.493 ± 0.022 |
| FCTalker [19] | 0.741 | 65.241 | 3.405 ± 0.026 | 3.537 ± 0.016 | 0.756 | 65.375 | 3.490 ± 0.029 | 3.491 ± 0.023 |
| ECSS [34] | 0.749 | 64.564 | 3.597 ± 0.024 | 3.585 ± 0.028 | 0.761 | 63.654 | 3.507 ± 0.018 | 3.587 ± 0.031 |
| **GPT-Talker (Ours)** | **0.882** | **42.125** | **3.890 ± 0.033** | **3.908 ± 0.029** | **0.884** | **45.627** | **3.884 ± 0.031** | **3.891 ± 0.017** |
| **Ground Truth** | - | - | **4.486 ± 0.026** | **4.501 ± 0.025** | - | - | **4.399 ± 0.020** | **4.493 ± 0.027** |

**Table 4: Subjective and objective experimental results on the analysis of three-stage training. (∗ means suboptimal result.)**

| Training Strategy | NCSSD (EN) | | | | NCSSD (ZH) | | | |
|---|---|---|---|---|---|---|---|---|
| | SSIM (↑) | DTWD (↓) | N-DMOS (↑) | E-DMOS (↑) | SSIM (↑) | DTWD (↓) | N-DMOS (↑) | E-DMOS (↑) |
| One-Stage (*w*/ CL & RC) | 0.843 | 57.734 | 3.609 ± 0.024 | 3.698 ± 0.014 | 0.851 | 56.842 | 3.625 ± 0.026 | 3.731 ± 0.021 |
| Two-Stage (*w*/ CL) | 0.875 | 48.653 | 3.713 ± 0.021 | 3.714 ± 0.017 | 0.877 | 47.854 | 3.716 ± 0.018 | 3.690 ± 0.030 |
| Two-Stage (*w*/ RC) | 0.879 | 47.863 | 3.716 ± 0.027 | 3.781 ± 0.023 | 0.884 | 48.125 | 3.696 ± 0.018 | 3.709 ± 0.029 |
| Two-Stage (*w*/ CL & RC) | 0.888 | 44.834 | 3.902 ± 0.033 | **3.925 ± 0.028** | 0.891 | 44.682 | **3.910 ± 0.030** | 3.916 ± 0.032 |
| **Three-Stage (Ours)** | **0.904** | **42.076** | **3.910 ± 0.019** | $3.922 ± 0.022^{*}$ | **0.908** | **43.002** | $3.906 ± 0.020^{*}$ | **3.987 ± 0.021** |

and E-DMOS based on the guidelines. Their compensation was calculated based on the number of test samples. Additionally, we calculate SSIM and DTWD, for the 50 test samples. The results for all metrics are presented in Table 3.

From the DTWD results in the table, it is evident that the speech synthesized by GPT-Talker outperforms the baselines, indicating that GPT-Talker is more capable of capturing pitch-related expressiveness similar to the GT. Furthermore, the N-DMOS and E-DMOS results show that our GPT-Talker significantly outperforms all the baselines. For example, in the DailyTalk dataset, GPT-Talker achieves N-DMOS and E-DMOS scores of 3.890 and 3.908, respectively, while the baselines score below 3.6. Similar trends are observed in the results for the NCSSD dataset. Moreover, GPT-Talker exhibits closer resemblance to GT compared to the baselines. These results demonstrate that GPT-Talker, leveraging ConGPT, effectively models the impact of context on the semantics and style of the current sentence, while ConVITS successfully renders semantic and stylistic aspects in CSS, resulting in highly natural and expressive synthesized speech. Lastly, by observing the SSIM metric results, it can be seen that GPT-Talker achieves the highest speaker similarity compared to all the baselines, proving that our ConVITS excels in timbre rendering on top of semantic and stylistic rendering. With the dialogue context modeling ability of ConGPT, ConVITS achieves highly expressive conversational speech synthesis.

### 6.3 Analysis of Three-Stage Training

This section validates the three-stage strategy, introduced in Section 3.4, used to train GPT-Talker. We design the following four training strategies for validation: 1) **One-Stage (*w*/ CL&RC)** means we directly use both the collection and recording subsets of NCSSD to train the GPT-Talker; 2) **Two-Stage (*w*/ CL&RC)** means we follow the same method as described in section 3.4 for pre-training. Afterward, we merge the two subsets together and perform fine-tuning; 3) **Two-Stage (*w*/ CL)**. This method is similar to the second one, with the difference that during fine-tuning, only the collection subset is selected; 4) **Two-Stage (*w*/ RC)**: This method is similar to

the second one, with the difference that during fine-tuning, only the recording subset is selected. We conduct subjective and objective experiments using the same configuration as in the previous section and report all results in Table 4. We observe that the multi-stage approach outperforms the single-stage approach in all metrics, suggesting that using a pre-training strategy enables the model to learn basic speech generation capabilities, and fine-tuning on the conversational data helps improve dialogue context understanding and expression abilities. Additionally, the "Two-Stage (*w*/ CL&RC)" and our three-stage approaches outperform previous two-stage methods, demonstrating that fine-tuning on the entire conversational dataset allows the model to learn dialogue expression capabilities with the support of a large volume of data. Comparing the "Two-Stage (*w*/ CL&RC)" approach with our three-stage approach, although there is a slight lag in one metric for both the Chinese and English datasets, our three-stage approach exhibits significant advantages in the remaining six metrics. This confirms that our three-stage method gradually guides the model to learn richer dialogue expression information, leading to better conversational speech generation capabilities.

## 7 Conclusion

In this work, we propose a novel GPT-based conversational speech synthesis (CSS) model, termed GPT-Talker, for user-agent interaction. It consists of ConGPT and ConVITS to model the semantic and style expression in the unified user-agent discrete dialogue context sequence and infer the expressive speech for the agent. We also propose the largest-scale conversational speech synthesis dataset to date, termed NCSSD, which stands out in terms of language and data source diversity. This dataset can support the training of GPT-Talker and even future GPT-style CSS models, providing a valuable resource for advancing CSS technology. The comprehensive experiments are conducted on the reliability of the NCSSD and the effectiveness of our GPT-Talker. We encourage the research community to use the NCSSD for spoken dialogue modeling.

## 8 Acknowledgments

The research by Rui Liu was funded by the Young Scientists Fund of the National Natural Science Foundation of China (No. 62206136), and Guangdong Provincial Key Laboratory of Human Digital Twin (No. 2022B1212010004), and the "Inner Mongolia Science and Technology Achievement Transfer and Transformation Demonstration Zone, University Collaborative Innovation Base, and University Entrepreneurship Training Base" Construction Project (Supercomputing Power Project) (No.21300-231510). The research by Yifan Hu was funded by the Inner Mongolia University 2024 Graduate Student Research and Innovation Project (Key Project Grant No. 11200-5223737). The research by Haizhou Li was funded by National Natural Science Foundation of China (Grant No. 62271432), Shenzhen Science and Technology Program ZDSYS20230626091302006, and Shenzhen Science and Technology Research Fund (Fundamental Research Key Project Grant No. JCYJ20220818103001002).

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
