# OpenReview forum: "Generative Expressive Conversational Speech Synthesis"
_acmmm.org/ACMMM/2024/Conference — MM2024 Poster_

### Official Review · Reviewer_6QF4 · 2024-05-22

**Rating:** 4
**Confidence:** 3

**Summary:**

Existing CSS methods generate speech by designing complex network structures and meticulously optimizing the modules within them. Moreover, due to the small amount of data in existing datasets that contain scripted recording styles, it is difficult to simulate real, natural conversational styles. To address these challenges, this work proposes a novel generative expressive CSS system, named GPT-talker. Additionally, this work introduces a large-scale bilingual CSS dataset, NCSSD, for training GPT-talker, and achieving state-of-the-art results.

**Strengths:**

1. By tokenizing textual and acoustic input into GPT, generating discrete speech tokens, and then using the ConVITS to produce the final speech, the method is intuitive.
2. A natural conversational speech corpus containing 236 hours is proposed, which is beneficial for future research in this field.
3. After training on the proposed dataset with GPT-talker, the model outperforms baseline models in terms of naturalness and expressiveness.

**Limitations:**

1. Textual tokens and acoustic tokens belong to two different feature spaces, and a purely GPT structure may not align these two features well.
2. Why does using an alternating approach between the text and speech modalities better capture the contextual dependencies of a conversation? GPT relies on all previous information to predict the next token, and the alternating approach does not seem to change the input information.

**Suitability:**

3

---

### Official Review · Reviewer_prwU · 2024-06-04

**Rating:** 4
**Confidence:** 3

**Summary:**

This paper advances the task of conversational speech synthesis in two aspects. First, the authors proposed to apply GPT-style language modeling to consider the conversational context and generate the corresponding semantics and style. Second, the authors curated a new conversational speech synthesis dataset called NCSSD, which is by far the largest dataset for conversational speech synthesis.

**Strengths:**

I am convinced that GPT-style language models are indeed good fits for the task of conversational speech synthesis given their outstanding ability to consider long-term context. Open-sourcing NCSSD is also no doubt a huge contribution to the community.

**Limitations:**

I have several concerns regarding the methodology.

First, I am concerned about how much style information resides in the HuBERT indices. For instance, previous works like AudioLM use audio codecs like SoundStream, which contain much more variation and information, and it is much easier to imagine that such codecs contain the necessary variation. On the other hand, HuBERT indices are known to contain mostly linguistic information, thus this model choice is a bit strange to me.

Second, the novelty of the conversational VITS is not significant, as it does not differ much from conventional VITS. In addition, I am confused about the role of the "Agent's Semantic and Style" and "Reference Speaker for Agent". Is the goal to use some representation (here, HuBERT indices) to represent semantics and style, and use the latter to represent the speaker? Couldn't it be possible that the latter also contains style? Finally, why use VITS to map the tokens to speech, instead of using VALL-E-like models?

I also find the "Reliability verification of NCSSD" experiment in Sec. 6.1 insufficient. "Training a single-utterance VITS with NCSSD and showing that the output is of comparable quality to the ground truth" seems somewhat unrelated to "NCSSD is a reliable dataset for CSS". Specifically, I question whether there are enough emotion and expressivity variations in NCSSD. I suggest authors to provide more analyses as those done in the STUDIES and CALLS papers.

My final concerns are about the multi-stage training strategy. First, in Sec. 6.3, the authors stated: "We observe that the multi-stage approach outperforms the single-stage approach in all metrics, suggesting that using a pre-training strategy enables the model to learn basic speech generation capabilities," however, I wonder about the possibility that the model simply benefits from being trained on more data. In addition, in Table 4, the "Two-Stage (𝑤/ CL & RC)" and "Three-Stage (Ours)" have their N-DMOS scores' confidence interval overlap, which means there are no statistical differences in performance. It is therefore difficult to persuade me that training on CL and then RC is indeed better. In fact, I am not really convinced in the first place -- why do the authors propose to pre-train on CL and then fine-tune on RC? Do the authors believe that RC is better than CL? If so, how do we verify it?

There are also some unclear parts about the methodology:
(1) Are the ConGPT and Conversational VITS models trained in an end-to-end manner?
(2) In the Conversational VITS model, how are $U^t_N$ and $T^a_N$ integrated? Specifically, what happens if they are of different lengths?

**Suitability:**

2

---

### Official Review · Reviewer_mFmg · 2024-06-09

**Rating:** 4
**Confidence:** 3

**Summary:**

The paper introduces GPT-Talker, a novel generative expressive Conversational Speech Synthesis system. GPT-Talker uses GPT to transform multimodal dialogue history into discrete token sequences, integrating them to predict responses that include both semantic and style knowledge. The system then synthesizes expressive speech using conversation-enriched VITS. Additionally, the authors present a new large-scale dataset, NCSSD, featuring 236 hours of naturally recorded conversational speech in both Chinese and English. The dataset includes dialogues from TV shows and improvised styles. Experiments demonstrate that GPT-Talker outperforms existing CSS systems in naturalness and expressiveness. The code, dataset, and pre-trained model would be available online.

**Strengths:**

Although there aren't any novel solid technical contributions in terms of new deep learning models, the overall framework looks interesting. The paper's primary innovation lies in transforming multimodal information from dialogue history into discrete token sequences, which are then integrated to form a comprehensive representation of the user-agent dialogue context.

The experiments conducted demonstrate that GPT-Talker significantly outperforms existing state-of-the-art CSS systems. User studies assess the naturalness and expressiveness of the synthesized speech, and the paper also employs objective metrics for evaluation. The reliability of the NCSSD dataset is validated by demonstrating its impact on the performance of the GPT-Talker system.

The contribution to the field, particularly with the publicly available dataset is laudable.

**Limitations:**

The usage of timbre rendering modules is well-established within the community, as is the capability of the VITS architecture for generating natural speech. It would be valuable to see results on optimizing GPT for conversational context, potentially involving pre-training on general text followed by fine-tuning on dialogue data. The paper does not demonstrate the ability to learn dialogue-specific tokens. Introducing dialogue-specific tokens or embeddings could help the model distinguish between different conversational roles (e.g., speaker vs. listener) and types of dialogue acts (e.g., questions, responses, affirmations).

The processing pipeline, including multimodal tokenization, context integration, and expressive speech synthesis, might introduce latency in real-time applications. Furthermore, the performance of the GPT-Talker system relies heavily on large-scale datasets like NCSSD, which are resource-intensive to create and may not be readily available for all languages or domains.

The current implementation and dataset focus on Chinese and English. Extending support to additional languages and dialects would require significant effort in collecting and annotating large-scale conversational datasets. Each language might also necessitate specific adjustments to handle unique linguistic features and conversational norms. Additionally, the system may not fully capture cultural nuances that influence conversational styles and expressions, potentially affecting the naturalness and appropriateness of synthesized speech in diverse cultural contexts.

While the paper focuses on naturalness and expressiveness in its evaluations, other crucial factors such as robustness to noisy inputs, variability in speaking styles, and adaptability to different user preferences are not thoroughly explored. A comprehensive evaluation across a broader set of metrics is necessary to fully understand the system's strengths and weaknesses. The evaluations are conducted in controlled settings, which may not accurately reflect real-world deployment scenarios. Factors such as background noise, diverse user accents, and varying conversational dynamics can impact the system’s performance in practical applications.

**Suitability:**

3

---

### Meta-Review · Area_Chair_xbbk · 2024-07-02

**Recommendation:** Accept (Poster)
**Confidence:** 3

**Metareview:**

This paper proposes a novel generative expressive Conversational Speech Synthesis system, GPT-Talker. The system creates dialogue token sequences from multimodal multi-turn dialogue history utterances, that are then posteriorly provided to an LLM as dialogue context. Then, the LLM is used to generate responses that cover both the style and semantics. Finally, generations are synthesized with a conversational VITS.

The paper got three positive (Borderline Accept) reviews, indicating that while there are some limitations with the submitted work, the approach is relevant for the community.

As key strengths, reviewers highlight the general soundness of the framework (mFmg, prwU, 6QF4), the NCSSD dataset contribution (mFmg, prwU, 6QF4), and the performance of the proposed approach (mFmg, 6QF4).

However, some limitations were also pointed out, such as:
- The lack of dialogue-specific tokens limits the expressiveness of the model (mFmg)
- An assessment of Acoustic Tokenizers is missing (prwU)
- An emotion and expressivity variations assessment, on the NCSSD dataset, is not fully clarified (prwU)
- The alternating concatenation effectiveness is only partially confirmed (6QF4)

The author's rebuttal only partially addressed reviewers' initial concerns, with all reviewers having maintained their initial scores.

Striking the balance between the strengths and limitations of the paper, I suggest this work to be accepted as Poster, if there is space.

Some of the pointed limitations hint at the relevance and research potential of the proposed work, and could be addressed in future work, such as a) latency optimization, b) expanding the framework to other languages, and c) assessment of adaptability to other factors (e.g. different speaking styles).